# Theoretical Kinetic Study of Thermal Decomposition of 5-Methyl-2-ethylfuran

**DOI:** 10.3390/molecules30071595

**Published:** 2025-04-02

**Authors:** Wei He, Cheng Wang, Qichuan Zhang, Kaixuan Chen, Linghao Shen, Yan Li, Kang Shen

**Affiliations:** 1Eastern Michigan Joint College of Engineering, Beibu Gulf University, Qinzhou 535011, China; hwei@bbgu.edu.cn (W.H.); slh675495142@163.com (L.S.); 2Guangxi Key Laboratory of Ocean Engineering Equipment and Technology, Qinzhou 535011, China; 15255117818@163.com (C.W.); zqc13355362749@163.com (Q.Z.); chenkx@bbgu.edu.cn (K.C.); 18176252822@163.com (Y.L.); 3Key Laboratory of Beibu Gulf Offshore Engineering Equipment and Technology, Beibu Gulf University, Education Department of Guangxi Zhuang Autonomous Region, Qinzhou 535011, China; 4College of Electrical Engineering, Guangxi University, Nanning 530004, China

**Keywords:** 5-Methyl-2-ethylfuran, pyrolysis, theoretical calculations, rate constant

## Abstract

With the advancement of new synthetic techniques, 5-Methyl-2-ethylfuran (5-MEF) has emerged as a promising renewable biofuel. In this study, the potential energy surfaces for the unimolecular dissociation reaction, H-addition reaction, and H-abstraction reaction of 5-MEF were mapped at the CBS-QB3 level. The temperature- and pressure-dependent rate constants for these reactions on the potential energy surfaces were determined by solving the master equation, using both transition state theory and Rice–Ramsperger–Kassel–Marcus theory. The results showed that the dissociation reaction of the C(6) site on the branched chain of 5-MEF has the largest rate constant and is the main decomposition pathway, while the dissociation reaction of the H atom on the furan ring has a lower rate constant and is not the main reaction pathway. In addition, the dissociation of H atoms on the branched chain and intramolecular H-transfer reactions also have high-rate constants and play an important role in the decomposition of 5-MEF. H-addition reactions mainly occur at the C(2) and C(5) sites, and the generation of the corresponding products through *β*-breakage becomes the main reaction pathway. With the increase in temperature, the H-addition reaction at the C(2) site gradually changes to a substitution reaction, dominating the formation of C_2_H_5_ and 2-methylfuran.

## 1. Introduction

Furan-based biomass fuels are a new type of renewable energy that is derived from the conversion of biomass materials (e.g., cellulose, hemicellulose) [1]. With advances in catalytic technology, biomass conversion processes, and green chemistry methods, the synthesis process has become more efficient and environmentally friendly [2,3]. These technological breakthroughs have not only improved the calorific value and combustion performance of the fuels, but also enhanced the sustainability of the industry, facilitated the commercialization of furan-based biomass fuels, and further promoted the development of clean energy [4,5]. However, the combustion characteristics of furan-based biomass fuels need to be studied before their actual use is crucial, and these studies can help optimize combustion efficiency and reduce pollutant generation [6]. These studies have focused on the combustion properties, pyrolysis mechanism, kinetic behavior, and pyrolysis mechanism of furans, 2-methyl furan, 2,5-dimethyl furan, 2-acetylfuran, methyl 2-furoate, 2-ethyl furan, and kinetic modeling.

Eldeeb et al. [7] measured the ignition delay time of 2,5-dimethylfuran and iso-octane in shock tubes and rapid compression machines, and the results of their study showed that DMF had a longer ignition delay time than iso-octane at lower temperatures. Somers et al. [8] systematically investigated the pyrolysis and oxidation properties of 2,5-dimethylfuran (DMF) by combining experiments with chemical kinetics simulations. The experiments were carried out using a flow reactor or a surge tube, combined with GC/MS and other assays, to analyze the decomposition pathways and product distributions (e.g., CO, CH_4_, C_2_H_4_, etc.) of DMF under different conditions, and combustion parameters, such as ignition delays and flame velocities, were measured. The kinetic model was constructed based on quantum chemical calculations, and the sensitivity analysis revealed that DMF was mainly consumed through furan ring-opening and H abstraction reactions, and its oxidizing activity was higher than that of conventional hydrocarbon fuels. Tanoue et al. [9] investigated the effect of furan and nitromethane additions to hydrocarbon fuels on combustion characteristics and showed that the addition of nitromethane lowered ignition temperatures and reduced carbon monoxide emissions, but may have increased nitrogen oxides, while furans contributed to a reduction in carbon soot production, although they may have increased formaldehyde emissions. Whelan et al. [10] investigated the reaction kinetics between hydroxyl radicals (OH) and furans and their alkylated derivatives (2-methylfuran and 2,5-dimethylfuran) by theoretical calculations, and the results of the study showed that 2,5-dimethylfuran exhibited the highest reactivity, suggesting that the addition of alkyl groups further accelerates the decomposition process. Pintor et al. [11] investigated the effect of the stability of 2-methylfuran and 2,5-dimethylfuran on the auto-ignition and combustion characteristics of standard gasoline fuels, and showed that the auto-ignition delay was longer for 2,5-dimethylfuran. Wu et al. [12] developed a compact and reliable kinetic mechanism for simulating the combustion process of furan biomass fuels (e.g., furan, 2-methylfuran, and 2,5-dimethylfuran) in an internal combustion engine, and showed that the mechanism efficiently captured the combustion characteristics of the furan fuels, which led to more efficient prediction of the ignition behavior, flame propagation, and emission characteristics. Tranter et al. [13] studied high-temperature pyrolysis experiments of 2-methylfuran in a flow reactor, and the results showed that 2-methylfuran undergoes C-O bond cleavage, ring opening, and free-radical chain reactions in high-temperature pyrolysis, generating a wide range of small-molecule products, and the products are transformed to simpler molecules as the temperature rises. Xu et al. [14] measured the ignition delay time of 2,5-dimethylfuran in a rapid compression machine and compared the ignition delay data of 2,5-dimethylfuran with those of 2-methylfuran and furan, which showed that 2,5-dimethylfuran had the shortest ignition delay, followed by 2-methylfuran, and furan had the longest delay.

Wang et al. [15] carried out experiments on 2-furfuryl alcohol pyrolysis in a flow reactor and a jet-stirred reactor, constructed a detailed kinetic model to simulate the decomposition process of 2-furfuryl alcohol, and validated the model with experimental data, which showed that C-O bond dissociation and H-abstraction reactions are the main decomposition pathways for furan consumption. Li et al. [16] carried out oxidation experiments of 2-ethylfuran in a jet-stirred reactor, and used synchrotron radiation vacuum ultraviolet photoionization mass spectrometry to identify and measure a variety of intermediates and products generated by the reaction, and developed a detailed kinetic model, which was validated by experimental data. The results showed that the consumption of 2-ethylfuran during the low-temperature oxidation of 2-ethylfuran was mainly due to the generation of 1-(2-furyl) ethyl radicals by hydrogen atom abstraction and 2, 3-dihydro-2-hydroxy-2-ethyl-3-furyl radicals by addition of hydroxyl radicals (OH) to C(2) of 2-ethylfuran. Su et al. [17] carried out experiments on the pyrolysis of 3-methylfuran in a flow tube and used synchrotron vacuum UV-ionization mass spectrometry to identify and measure a variety of intermediates and products generated from the reaction, and detected and measured key products and intermediates including methyl, propargyl, ethynyl, ethylene, propargyl/propadienyl, vinyl ethynyl, propene, 1,3-butadiene, 2-butynyl, ethyl ketone, and so on. Finally, a pyrolysis model of MF3 was constructed and validated against experimental data. He et al. [18] carried out pyrolysis experiments of 2-acetylfuran in a jet stirred reactor (JSR) setup using vacuum ultraviolet photoionization mass spectrometry (SVUV-PIMS) with molecular beam sampling and gas chromatography. Based on the combustion model of 2-methylfuran, a detailed pyrolysis model of 2-acetylfuran was constructed by theoretical calculations and analogies, and the model was validated by current pyrolysis experiments. The rate of production (ROP) analysis indicates that 2-acetylfuran consumption is mainly controlled by single-molecule dissociation reactions and H-addition reactions. Yan et al. [19] investigated the pyrolysis experiments of methyl 2-furoate in a flow reactor and constructed the potential energy surface for the monomolecular dissociation reaction of Methyl 2-furoate at the level of CBS-QB3, and established the first pyrolysis model of Methyl 2-furoate, which was verified experimentally. The ROP analysis shows that the consumption during Methyl 2-furoate pyrolysis mainly includes monomolecular dissociation reaction, in situ substitution reaction, H-addition reaction, and H-abstraction reaction.

He et al. [20] constructed potential energy surfaces for the unimolecular dissociation, H-addition and H-extraction reactions of 2-vinylfuran at the level of G4, and calculated the temperature- and pressure-dependent rate constants for the relevant reactions on the potential energy surfaces. The results show that the highest rate of constants was obtained for the intramolecular transfer of H atoms from the C(5) site to the C(4) site of 2-vinylfuran to form 2-vinylfuran-3(2H)-carbene, which then decomposes to form h145te3o.

5-Methyl-2-ethylfuran (5-MEF), as a potential biomass-derived fuel, has the advantages of high energy density, high octane number, excellent combustion performance, and low pollutant emission, which can be obtained through the conversion of sugars in biomass (e.g., lignocellulose). Before the practical application of 5-MEF as a potential biomass fuel, it is crucial to systematically study its combustion characteristics. As a novel fuel candidate, the combustion behavior of 5-MEF will directly affect its suitability and efficiency in combustion systems such as internal combustion engines. The research on 5-MEF as a biomass fuel is still in its infancy, and there are few studies related to its combustion characteristics.

In this work, the high-temperature pyrolysis mechanism of 5-MEF was systematically constructed by theoretical calculations (including the determination of reaction pathways, analysis of intermediates, and calculation of rate constants, etc.) to gain a deeper understanding of its decomposition pathways and the generation of intermediates under high-temperature conditions.

## 2. Results and Discussion

### 2.1. Potential Energy Surfaces

In this study, the potential energy surfaces of the 5-MEF reaction system were calculated at the level of CBS-QB_3_, including unimolecular dissociation, H-transfer decomposition, H-abstraction, and H-addition. These calculations provide theoretical guidance for understanding the reaction pathways and associated energy barriers for each reaction pathway. The bond dissociation energy is a key thermodynamic parameter in explaining and predicting chemical reaction pathways, reactivity, and molecular stability [21]. The molecular structure and bond dissociation energy of 5-MEF were calculated at the CBS-QB3 level as shown in Figure 1.

#### 2.1.1. Unimolecular Dissociation Reaction Potential Energy Surface of 5-MEF

Unimolecular decomposition of 5-MEF mainly involves the cleavage of C(8)-H, C(7)-H, C(6)-H, C(6)-CH_3_, C(2)-C_2_H_5_, and C(5)-CH_3_ bonds on the branched chain to form H + (5-ethylfuran-2-yl)methyl radical, H + 2-(5-methylfuran-2-yl)ethyl radicals, H + 1-(5-methylfuran-2-yl)ethyl radicals, CH_3_ + (5-methylfuran-2-yl)methyl radicals, C_2_H_5_ + 5-methylfuran-2-yl radical, and CH_3_ + 5-ethylfuran-2-yl radical with energy barriers of 84.3, 101.2, 81.9, 72.0, 111.3, and 84.3 kcal·mol^−1^, respectively. Energetically, the H atom at the C(6) site on the branched chain is most susceptible to dissociation reactions.

Additionally, this study calculates the H-dissociation reactions on the furan ring of 5-MEF, specifically for the C (3)-H and C (4)-H bonds. These reactions yield H + 2-ethyl-5-methylfuran-3-yl and H + 5-ethyl-2-methylfuran-3-yl with energy barriers of 118.4 and 118.5 kcal·mol^−1^, respectively. The bond dissociation energies for 5-MEF, as depicted in Figure 2, also indicate that H-dissociation on the 5-MEF branched chain is energetically more favorable compared to H-dissociation on the furan ring.

#### 2.1.2. H-Transfer Decompositions Potential Energy Surface of 5-MEF

The PES for H-transfer decomposition in furan highlights the intricate balance between stability, reactivity, and energy, crucial for controlling and optimizing chemical reactions involving furan and similar compounds.

Through H-transfer reaction, 5-MEF may be converted to 2-ethyl-5-methylfuran-3(2*H*) carbene and 5-ethyl-2-methylfuran-3(2*H*) carbene with the energy barriers 68.8 kcal·mol^−1^ (TS20) and 69.1 kcal·mol^−1^ (TS22), respectively. Subsequently, 2-ethyl-5-methylfuran-3(2*H*) carbene and 5-ethyl-2-methylfuran-3(2*H*) carbene undergo decomposition reactions to form hepta-3,4-dien-2-one and hepta-4,5-dien-3-one with energy barriers of 58.9 kcal·mol^−1^ (TS21) and 59.2 kcal·mol^−1^ (TS23), respectively. The energy barrier of the intramolecular H-transfer decomposition reaction of 5-MEF is more dominant than that of the unimolecular dissociation reaction, which dominates the beginning of the fuel pyrolysis reaction.

#### 2.1.3. H-Abstraction Reaction Potential Energy Surface of 5-MEF

In the context of combustion, H-abstraction is often initiated by radicals (such as OH, O, or H) that react with furan to create a reactive furan radical [20]. This step is essential in triggering chain reactions that ultimately lead to furan’s decomposition and complete combustion into CO_2_, H_2_O, and other smaller molecules.

Figure 3 and Figure 4 show the potential energy surfaces for the H-abstraction reactions of 5-MEF with the H atom, and 5-MEF with the CH_3_, respectively. Two H-atom abstraction sites on the branched ethyl chain of 5-MEF react with H atoms to form 2-(5-methylfuran-2-yl)ethyl radical + H_2_ and 1-(5-methylfuran-2-yl)ethyl radical + H_2_ with energy barriers of 10.8 kcal·mol^−1^ (TS3) and 4.2 kcal·mol^−1^ (TS2), respectively. In addition, the CH_3_ in the branched chain of 5-MEF reacts with H atoms in an H-abstraction reaction to form (5-ethylfuran-2-yl)methyl radical + H_2_ with an energy barrier of 5.9 kcal·mol^−1^ (TS1). H atoms on C(3) and C(4) of the 5-MEF ring undergo H-abstraction reactions with H atoms to form 2-ethyl-5-methylfuran-3-yl radical + H_2_ and 5-ethyl-2-methylfuran-3-yl radical + H_2_ with energy barriers of 19.9 kcal·mol^−1^ (TS4) and 19.7 kcal·mol^−1^ (TS5), respectively.

The H-abstraction reaction of methyl groups with 5-MEF was also considered in the current work. There are two H-atom abstraction sites on the branched ethyl chain of 5-MEF that react with methyl to form 2-(5-methylfuran-2-yl)ethyl radical + CH_4_ and 1-(5-methylfuran-2-yl)ethyl radical + CH_4_ with energy barriers of 12.9 kcal·mol^−1^ (TS8) and 16.2 kcal·mol^−1^ (TS7), respectively. The H atom on the CH_3_ in the branched chain of 5-MEF undergoes a H-abstraction reaction with the CH_3_ to form (5-ethylfuran-2-yl)methyl radical + CH_4_ with an energy barrier of 9.2 kcal·mol^−1^. The H atoms on C (3) and C (4) of the 5-MEF ring undergo H-abstraction reactions with methyl groups to form 2-ethyl-5-methylfuran-3-yl radical + CH_4_ and 5-ethyl-2-methylfuran-3-yl radical + CH_4_ with energy barriers of 19.3 kcal·mol^−1^ (TS9) and 19.3 kcal·mol^−1^ (TS10), respectively. It is interesting to note that in the current calculations the difference between the energy barriers and the relative energies of the products of the H-abstraction reactions of CH_3_ with H atoms on the 5-MEF ring is very small.

The methylidene site on the ethyl group of the 5-MEF branch is the most competitive in terms of energy barrier for the H-abstraction reaction. The H atoms on the furan ring of 5-MEF have a higher energy barrier for the inverse H-abstraction reaction and the reaction channel is not dominant, as confirmed by the bond dissociation energies in Figure 1.

#### 2.1.4. H-Addition Reaction Potential Energy Surface of 5-MEF

In combustion, H-addition reactions are critical in shaping the reaction pathways, altering stability, and ultimately affecting product formation. This process generally involves radical species, where an H atom adds to a specific site on the furan ring, potentially leading to subsequent reactions that facilitate further decomposition and oxidation [20].

The addition reactions of four sites on the 5-MEF ring with H atoms were calculated at the level of CBS-QB3 as shown in Figure 5. H-addition to the 5-MEF C (2) generates 2-ethyl-5-methyl-2,3-dihydrofuran-3-yl radicals with energy barrier of 2. 0 kcal·mol^−1^(TS13). The 2-ethyl-5-methyl-2,3-dihydrofuran-3-yl radicals subsequently *β*-break the O(1)-C(2) and C(2)-C(6) bonds to produce (*E*)-2-oxohept-4-en-3-yl and C_2_H_5_ + 2-methylfuran with energy barriers of 21.1 kcal·mol^−1^ (TS15) and 29.5 kcal·mol^−1^ (TS14), respectively. The 5-MEF requires less energy to break the O (1)-C (2) bond than the C(2)-C(6) bond, so the reaction is more likely to occur. H-addition of C(3) to the 5-MEF ring generates the 2-ethyl-5-methyl-2,3-dihydrofuran-2-yl radical with an energy barrier of 3.5 kcal·mol^−1^ (TS11). Then, 2-ethyl-5-methyl-2,3-dihydrofuran-2-yl radical generates the (*Z*)-5-oxohept-2-en-2-yl radical via the *β*-break O(1)-C(5), with an energy barrier of 33.8 kcal·mol^−1^ (TS12). The energy required for this reaction channel is too high, resulting in a reaction channel that does not easily occur. H-addition of C(4) to the 5-MEF ring leads to the 5-ethyl-2-methyl-2,3-dihydrofuran-2-yl radical with an energy barrier of 3. 6 kcal·mol^−1^ (TS16). Subsequent generation of 5-ethyl-2-methyl-2,3-dihydrofuran-2-yl radical via *β*-break O(1)-C(2) generates (*E*)-6-oxohept-4-en-3-yl radical at an energy barrier of 13.0 kcal·mol^−1^ (TS17). H-addition at the 5-MEF ring of C(5) generates the 5-ethyl-2-methyl-2,3-dihydrofuran-3-yl radical with an energy barrier of 2. 3 kcal·mol^−1^ (TS18). Subsequent production of 5-ethyl-2-methyl-2,3-dihydrofuran-3-yl via *β*-break O(1)-C(5) generates the (*Z*)-5-oxohept-2-en-4-yl radical with an energy barrier of 21.1 kcal·mol^−1^ (TS18).

Analysis on the potential energy surface shows that the reaction channel for H-addition to the 5-MEF ring C(5) is the most energetically competitive.

### 2.2. Temperature and Pressure-Dependent Rate Coefficients

The temperature- and pressure-dependent rate coefficients in furan combustion provide insight into the behavior of elementary steps and help optimize conditions for efficient energy release and minimal pollutant formation.

#### 2.2.1. Unimolecular Dissociation and H-Transfer Decomposition Reaction Rate Constants for 5-MEF

The rate constants for both the unimolecular dissociation and H-transfer decomposition reactions of 5-MEF are dependent on temperature and pressure. The rate constants for both types of reactions generally increase with increasing temperature because higher temperatures provide more energy to overcome the activation energy barrier. Unimolecular dissociation reactions are more pronounced at higher temperatures, while hydrogen transfer reactions dominate at lower temperatures.

As shown in Figure 6, the dissociation reaction at the C(6) site on the 5-MEF branched chain produces (5-methylfuran-2-yl)methyl radicals and CH_3_ with the largest rate constants, making it the major reaction channel. In contrast, the rate constant for the dissociation reaction of the hydrogen atom on the 5-MEF furan ring is significantly lower than that on the branched chain by several orders of magnitude and thus is not the major decomposition channel. In addition, the H atom dissociation reaction on the 5-MEF branched chain and the intramolecular H-transfer decomposition reaction also have high-rate constants, and they are also important reaction channels in the decomposition of 5-MEF.

The rate constants for both the single-molecule dissociation reaction and the H-transfer decomposition reaction of 5-MEF are temperature- and pressure-dependent. The rate constants for both types of reactions generally increase with increasing temperature because higher temperatures provide more energy to overcome the activation energy barrier. Unimolecular dissociation reactions are more pronounced at higher temperatures, while H-transfer reactions dominate at lower temperatures.

#### 2.2.2. H-Abstraction Reaction Rate Constants for 5-MEF

Figure 7 illustrates the rate constant for the H-abstraction reaction of 5-MEF with H atoms. The results show that the C(6) site has the largest rate constant for H-abstraction, followed by the reaction channel at the C(8) site. In contrast, the rate constants of H-abstraction reactions at the C(3) and C(4) sites on the furan ring were smaller and significantly lower than those of the reaction channels at the branched positions, and thus they were not the major reaction channels. In addition, the difference in the H-abstraction reaction rate constants between the C(6) and C(7) sites was large at temperatures lower than 1100 K, but the gap between the rate constants of the two reaction channels gradually narrowed as the temperature increased (see Appendix A).

Figure 8 shows the rate constant for the abstraction reaction of 5-MEF with CH_3_. Similar to the H atom abstraction reaction, the C(6) site has the largest H-abstraction rate constant, followed closely by the reaction channel at the C(8) site. In contrast, the H-abstraction reaction rate constants for the C(3) and C(4) sites on the furan ring were smaller and significantly lower than those of the reaction channels at the branched positions, and thus they were not the major reaction pathways. Further, the difference in rate constants between the C(6) and C(7) sites was large at temperatures lower than 1100 K, but the gap between the rate constants of these two reaction channels gradually narrowed as the temperature increased.

#### 2.2.3. H-Addition Reaction Rate Constants for 5-MEF

H-addition reactions usually occur faster at higher temperatures because higher temperatures increase the frequency of collisions between molecules, thus providing more energy to overcome the activation energy of the reaction. In 5-MEF molecules, C=C double bonds are the most common reaction sites and H atoms may be added to these double bonds. The addition of H atoms to the C(2) and C(5) sites is followed by *β*-breakage reactions to generate the corresponding products, and these two reaction channels have the largest rate constants and thus become the major reaction pathways, as shown in Figure 9. Under low-pressure conditions, the H-addition at the C(2) site mainly generates the corresponding products via *β*-breakage. With increasing temperature, the H-addition reaction at the C(2) site gradually changes to a substitution reaction, dominating to produce C_2_H_5_ and 2-methylfuran. The rate constants for the addition of H atoms to the C(3) and C(4) sites are several orders of magnitude smaller than those for the C(2) and C(5) sites at low and high pressures, making these two reaction paths uncompetitive.

## 3. Theoretical and Computational Methods

### 3.1. Electronic Structure Methods

Geometry optimization, zero-point energy, and frequency analysis of all stationary points on the 5-MEF potential energy surface using the B3LYP/6-311G(2*d*, *d*, *p*) method [22]. This is verified by inspection of the hindered rotor potential to ensure that the geometry of the stationary point on the potential energy surface is the lowest energy structure. Transition states were identified by having one and only one imaginary frequency and verified by vibrational modes to correspond to the desired response coordinates. For uncertainty, intrinsic reaction coordinate analysis was performed to ensure that the transition state was connected to the indicated reactants and products [23]. The more accurate relative energies of these species were obtained at the CBS-QB_3_ (0 K) level, which guarantees an energy precision within the range of approximately 1 kcal·mol^−1^ [24]. All density functional calculations were performed using the Gaussian 09 program [25].

### 3.2. Reaction Kinetics

The temperature- and pressure-dependent rate constants for the corresponding reactions on the potential energy surface are calculated by solving the master equation based on the transition state theory and the Rice–Ramsperger–Kassel–Marcus theory with the Eckart [26] tunneling effect correction via the MESS program [27,28].

Mess programs have been developed to deal with complex systems including multi-potential wells and multi-reaction channels to obtain temperature- and pressure-dependent rate coefficients [29]. The traditional transition state theory is used to predict rate constants for reactions with significant energy barriers. The torsional motion of species and transition states treated with a one-dimensional hindered rotor is employed. Relaxation scans along dihedral coordinates at 10° intervals at the B3LYP/6-311G(2*d*, *d*, *p*) level were obtained for the hindered potential of the internal rotors. For the dissociated channel with no barrier, the rate constants were calculated using variational transition state theory. Due to the small rotational potential barrier, the torsional modes involved in the configuration of the transition state region were treated with a one-dimensional free rotor.

The interaction between the reactants and the bath gas Ar is represented by the Lennard-Jones (L-J) collision model. The L-J parameters for Ar are *σ* = 3.47 Å and *ε* = 79.2 cm^−1^, respectively, as recommended by Date [30]. The collision energy transfer probability is usually approximated using a single-exponential down model, i.e., <△*E*_down_> = 200 × (T/300)^0.75^ cm^−1^ [31] It was calculated using Joback’s group contribution scheme for the 5-MEF molecule with *σ* = 5.80 Å and *ε* = 340.63 cm^−1^ [32].

## 4. Conclusions

This study explores the pyrolysis potential energy surface of 5-Methyl-2-ethylfuran, which is considered a potential biomass fuel, at the CBS-QB3 level. The investigation includes unimolecular dissociation reactions, intermolecular H-transfer reactions, H-abstraction reactions, and H-addition reactions. Subsequently, the rate constants for the reaction pathways on the potential energy surface were determined by solving the master equation using RRKM theory and transition state theory. The study also discusses the influence of temperature and pressure on the rate constants.

The results showed that the dissociation reaction at the C(6) site on the branched chain of 5-MEF had the largest rate constant and was the main decomposition pathway, while the dissociation reaction of the H atoms on the furan ring had a significantly lower rate constant and was therefore not the main reaction pathway. In addition, the dissociation of H atoms on the branched chain and intramolecular H-transfer reactions also have high-rate constants and play an important role in the decomposition of 5-MEF. For H-abstraction reactions, the C(6) site has the largest rate constant, followed by the C(8) site, while the C(3) and C(4) sites have smaller rate constants and are not the major reaction pathways. H-addition reactions mainly occurred at C(2) and C(5) sites and generated the corresponding products through *β*-breakage, which became the main reaction pathway. With the increase in temperature, the H-addition reaction at the C(2) site gradually changed to a substitution reaction, dominating the generation of C_2_H_5_ and 2-methylfuran. Overall, the reaction channels at the branched site are more competitive than those on the furan ring. This study provides detailed kinetic data on the reaction mechanism of 5-MEF and provides a theoretical basis for its application as a potential biofuel.

## Figures and Tables

**Figure 1 molecules-30-01595-f001:**
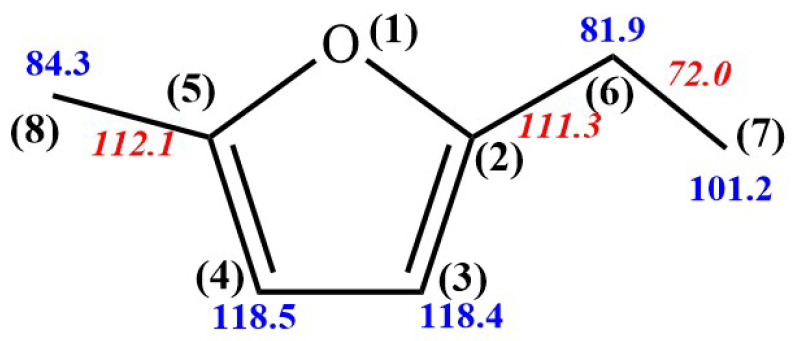
Calculate the molecular structure and bond dissociation energy (in kcal·mol^−1^) of 5−MEF at the CBS-QB3 level. Normal text, C–H bond; italic, C–C bond. Numbers in parentheses are atom labels. Zero-point energies are included.

**Figure 2 molecules-30-01595-f002:**
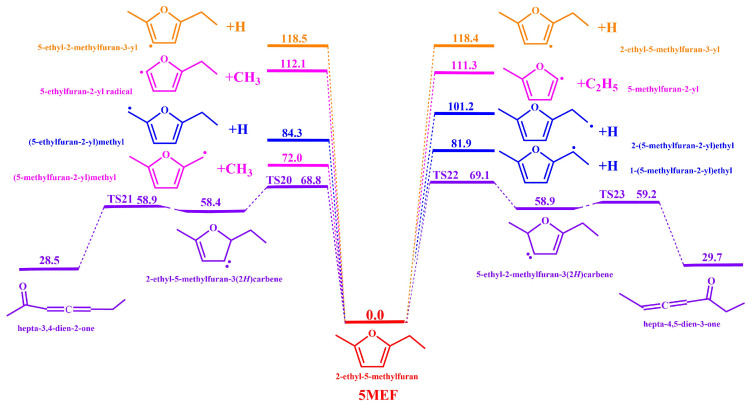
Potential energy surfaces calculated at the CBS-QB3 level for the major unimolecular decomposition pathways of 5−MEF. The unit is in kcal·mol^−1^. Zero-point energies are included.

**Figure 3 molecules-30-01595-f003:**
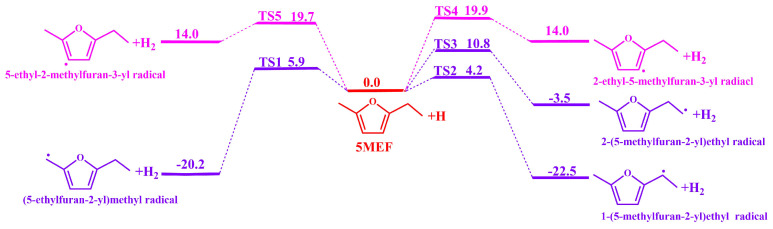
The potential energy surface for the H-abstraction reaction of H atoms with 5−MEF was calculated at the level of CBS-QB3. The unit is in kcal·mol^−1^. Zero-point energies are included.

**Figure 4 molecules-30-01595-f004:**
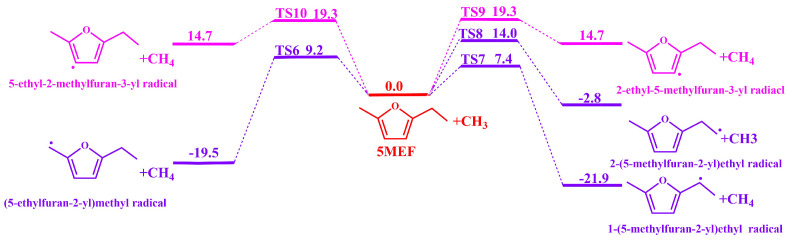
The potential energy surface for the H-abstraction reaction of CH_3_ with 5−MEF was calculated at the level of CBS-QB3. The unit is in kcal·mol^−1^. Zero-point energies are included.

**Figure 5 molecules-30-01595-f005:**
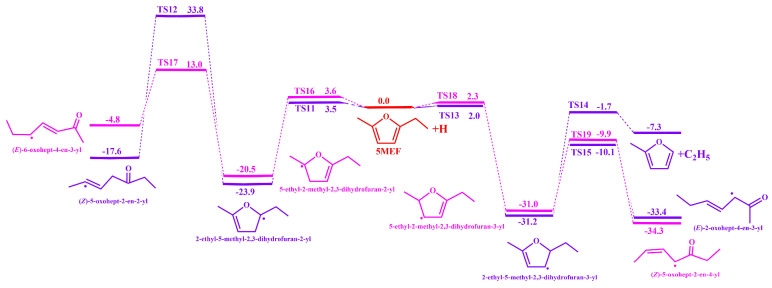
The potential energy surface for the addition of H atoms to 5−MEF was calculated at the CBS-QB3 level. The unit is in kcal·mol^−1^. Zero-point energies are included.

**Figure 6 molecules-30-01595-f006:**
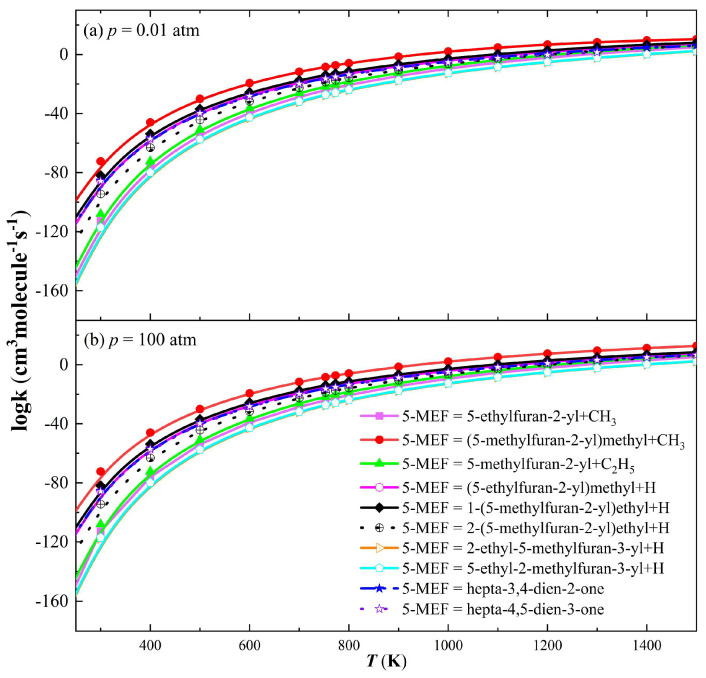
Rate constants for unimolecular dissociation and intramolecular H-transfer decomposition of 5−MEF. (**a**) *p* = 0.01 atm; (**b**) *p* = 100 atm.

**Figure 7 molecules-30-01595-f007:**
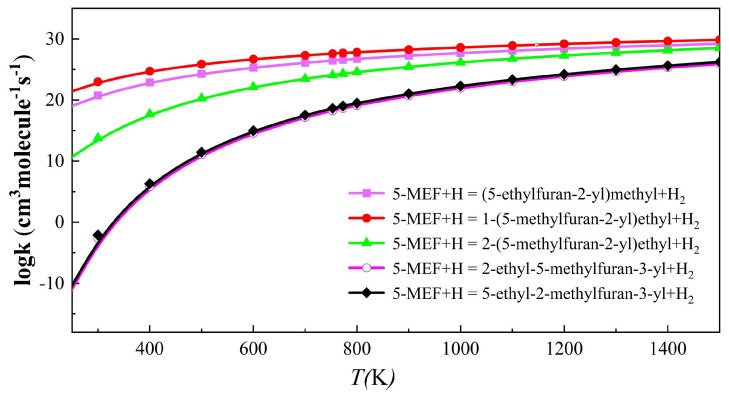
The rate constant for the H-abstraction reaction of 5−MEF with H at temperatures from 300 to 1500 K.

**Figure 8 molecules-30-01595-f008:**
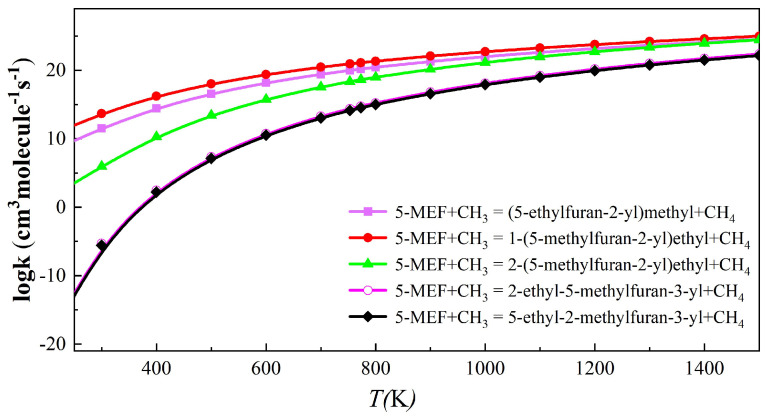
The rate constant for the H-abstraction reaction of 5-MEF with CH_3_ at temperatures from 300 to 1500 K.

**Figure 9 molecules-30-01595-f009:**
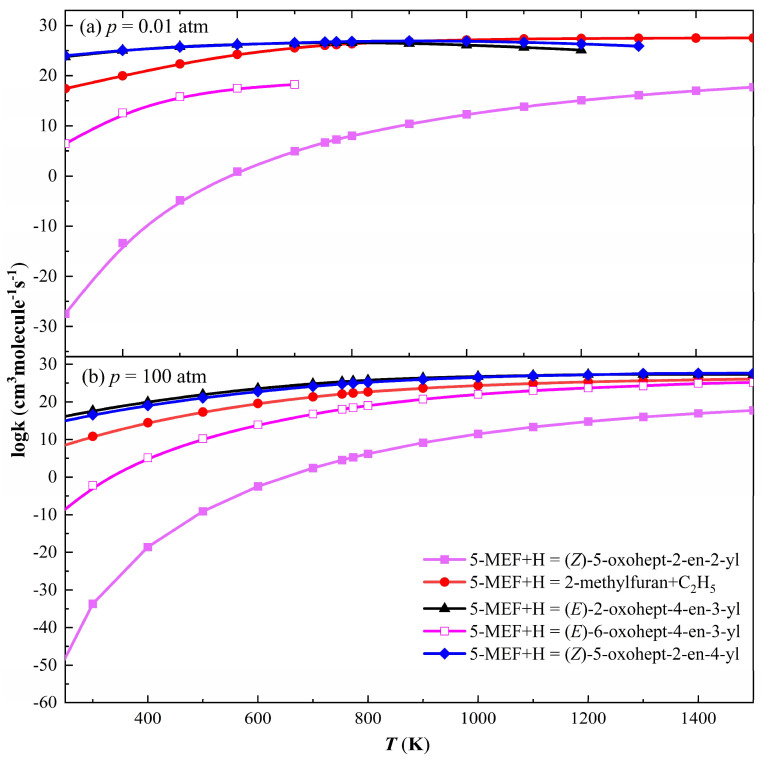
The rate constants for the addition reaction of 5-MEF with H at temperatures from 250 to 1500 K. (**a**) *p* = 0.01 atm; (**b**) *p* = 100 atm.

## Data Availability

The original contributions presented in this study are included in the article/Appendix A. Further inquiries can be directed to the corresponding author.

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
