# Peer review of "Theoretical Kinetic Study of Thermal Decomposition of 5-Methyl-2-ethylfuran"

_molecules, 2025, doi:10.3390/molecules30071595_

Round 1
Reviewer 1 Report
Comments and Suggestions for Authors
5-Methyl-2-ethylfuran (5-MEF) is considered a promising renewable biofuel. However, the combustion characteristics and reaction of its degradation must be studied. 5-MEF has advantages such as high energy density, high octane number, and low pollutant emission. This study aimes to investigate the high-temperature pyrolysis mechanism of 5-MEF using theoretical calculations to understand its decomposition pathways and kinetics under low-to high-pressure conditions and high temperatures.
The authors used the B3LYP DFT functional with the 6-311+G(2d,d,2p) basis set for geometry optimization and thermochemistry. CBS-QB3 calculations were used to improve energetics. The reaction kinetics were investigated using the RRKM (Rice-Ramsperger-Kassel-Marcus) approach to examine the effects of pressure. The level of computational calculation is sufficient to adequately characterize the reaction mechanism and kinetics because CBS-QB3 compensates for the tendency of B3LYP to underestimate the H transfer barriers. However, a few aspects need to be improved: the authors may have overlooked the diradical mechanisms and reactions initiated by OH and O.
The paper is interesting and well-written, and the results have been carefully analyzed. I recommend the publication of this manuscript after major revisions.
Major points:
- The title is misleading; this is not a comprehensive study of the decomposition of 5-Me- 2-ethyl-2-ethylfuran, because only the first steps of the oxidation were considered.
- Page 5, section 3.1.2. H-transfer from C8 to C5, C6 to C2, and C8 to C3 should also be considered. They form diradicals, which could be more stable than carbenes. Did the authors test these H transfers? In the case of diradicals, wavefunction stability should be checked.
- Page 6, section 3.1.3. The authors stated “H-abstraction is often initiated by radicals (such as OH, O, or H)” but the reactions initiated by O and OH were not considered.
Minor points:
- Page 4, line 150. Could the authors provide more details on how they obtained rate constants for barrierless reactions?
- Page 4, Figure 1. There is inconsistency between some numbers in Figure 1 and 2. The energies of C2-C6 and C6-C7 dissociations (111.3 and 72.0) in figure 1 seem to be inverted in Figure 2. Which are correct?
- Figures 1 to 5. The energies including the ZPE, should be reported instead of the potential energies.
- Page 5, line 180. “Energetically the H atom at the C(6) site on the branched chain is most susceptible to dissociation reactions.” According to the numbers in Figure 1, C(6)–C(7) is the bond that is most susceptible to dissociation.
- What is the electronic state of carbenes? Singlets or triplets?
- In the Introduction section, I recommend including a reference to the study by Somers et al. on the pyrolysis and oxidation of 2,5-dimethylfuran.
DOI: 1016/j.combustflame.2013.06.007 - For the sake of reproducibility, cartesian coordinates, vibrational frequencies and moments of inertia for all the structures should be reported in the Supplementary Material
Author Response
Dear Editor(s),
Thank you very much for your email and for the reviewers’ valuable comments and suggestions on our manuscript titled "Theoretical Kinetic Study of Thermal Decomposition of 5-Methyl-2-ethylfuran" (Manuscript ID: molecules-3529931).
We have carefully considered all the comments and have revised the manuscript accordingly. A detailed point-by-point response to each of the reviewers' comments is provided in the attached file, along with the revised version of the manuscript.
We greatly appreciate the reviewers’ time and constructive feedback, which helped us improve the quality of our work.
Please do not hesitate to contact us if any further information is needed.
Sincerely,
Kang Shen

Reviewer 2 Report
Comments and Suggestions for Authors
Authors should add more references and reduce the number of self-citations.
Author Response

(The authors gave the same response as above.)

Reviewer 3 Report
Comments and Suggestions for Authors
Report on: Theoretical Kinetic Study of Thermal Decomposition of 5-methyl-2-ethyl furan
This manuscript addresses the kinetics of some reactions involved in the combustion of a renewable biofuel. The methodology is quite adequate, but I would select another more appropriate functional for kinetic calculations, such as M06-2X. The basis set is sufficient, but I do not understand the nomenclature, 6-311G(2d, d, p). Maybe the authors mean 6-311G(2d, dp)? Why do not include diffusion functions?
Regarding the presentation, I suggest explaining or perhaps tabulating the role of entropy in the reaction rates. It is very different for unimolecular than for bimolecular reactions. Without that, the potential energy surfaces add little to the reader's understanding.
It would be nice to tabulate the rate equations, including temperature dependence. It is much more helpful than the plots of rate constants with temperature.
I recommend publication after these comments are addressed.
Author Response

(The authors gave the same response as above.)

Round 2
Reviewer 1 Report
Comments and Suggestions for Authors
The authors addressed the requests and questions raised in the first review. The paper is publishable in its present form.